# Efficacy of Listening to Music on Pain Reduction during Colposcopy-Directed Cervical Biopsy: A Randomized, Controlled Trial

**DOI:** 10.3390/medicina58030429

**Published:** 2022-03-16

**Authors:** Lalita Pradit, Charuwan Tantipalakorn, Kittipat Charoenkwan, Prapaporn Suprasert, Jatupol Srisomboon, Tanarat Muangmool

**Affiliations:** Department of Obstetrics and Gynecology, Faculty of Medicine, Chiang Mai University, Chiang Mai 50200, Thailand; lalita13pradit@gmail.com (L.P.); kittipat.c@cmu.ac.th (K.C.); prapaporn.su@cmu.ac.th (P.S.); jatupol1957@hotmail.com (J.S.); tanarat.m@cmu.ac.th (T.M.)

**Keywords:** cervical biopsy, colposcopy, music, pain relief

## Abstract

*Background and Objective:* To investigate the efficacy of listening to music on pain reduction during colposcopy-directed cervical biopsy (CDB). *Materials and Methods*: From June 2020 to November 2021, 240 women undergoing CDB were enrolled. The participants were randomized into three groups: Group 1, colposcopic examination while wearing headphones and listening to music; Group 2, colposcopy while wearing headphones but not listening to music; Group 3 (control group), colposcopy while neither listening to music nor wearing headphones. All participating women completed a 10 cm visual analog scale for subjective pain at three time points: baseline, immediately after cervical biopsy, and 15 min after the procedure. The primary endpoint was the biopsy pain score. *Result*: Of the 240 women, a sample size of 80 was randomly assigned per group. The clinical–pathological and procedure-related characteristics of the participants in all groups were similar. The mean baseline pain score between each group was not significantly different (2.83 in the music group, 2.54 in group 2, and 2.94 in the control group, *p* = 0.47). There were no significant differences between each group in terms of mean biopsy pain score (4.21 in the music group, 4.24 in group 2, and 4.30 in the control group, *p* = 0.98). The differences in changes between the baseline pain score and the biopsy pain score were not statistically significant (1.39 in the music group, 1.70 in group 2, and 1.36 in the control group, *p* = 0.69). In the multiple comparison analysis, the differences in changes between the biopsy pain score and the baseline pain score between each group were also not statistically significant. There were no complications with the intervention observed. *Conclusion*: This study demonstrated that there was no beneficial effect of listening to music on pain reduction during colposcopy-directed cervical biopsies.

## 1. Introduction

According to the 2020 global cancer statistics of the World Health Organization, an estimated 604,000 new cases of cervical cancer were diagnosed in 2020, accounting for 6.5 percent of all cancer cases in women [1]. In developing countries, of female cancers, cervical cancer was the second most common cancer and the third most common cause of cancer-related death [2,3,4]. In Thailand, cervical cancer is the third most common cancer, ranking after breast and colorectal cancer. There were 9157 new cervical cancer cases in Thai women, or 25 cervical cancer cases were diagnosed every day. In total, 4705 Thai women died from cervical cancer, or approximately 12 deaths each day. Accordingly, cervical cancer remains a leading health problem [1].

Colposcopy is an important procedure in cervical cancer prevention. It has the role in decreasing the incidence and mortality rates of cervical cancer. Colposcopy-directed cervical biopsy (CDB) is an outpatient diagnostic procedure performed in women with abnormal cervical cytology or positive human papillomavirus (HPV) testing [5]. The evidence-based practice guidelines recommend that biopsies be taken of abnormal acetowhite areas, generally two to four biopsies of colposcopic inspection [6]. However, such practice may cause the patients more discomfort, burning sensations, and pain associated with the biopsy. During the examination, additional intervention may be performed including endocervical brushing (ECB), endocervical curettage (ECC), endocervical polypectomy, and endometrial aspiration biopsy for further evaluation. This might also cause fear and anxiety throughout the examination. These factors lead to loss to follow-up and treatment [7]. Previous studies have tried to find methods to reduce pain and discomfort during colposcopy to improve treatment outcomes such as lidocaine injection, lidocaine spray, or forced coughing [8,9,10,11]. Music has been investigated in chronic pain, pain reduction in children, and other procedures such as orthopedic surgery or cystoscopy with promising outcomes [12,13,14,15]. Several studies have recently been conducted that addressed the effect of listening to music on pain and discomfort reduction during colposcopy [16,17,18,19,20,21]. Some studies indicated that listening to music significantly reduced pain and discomfort during colposcopy [16,17], whereas others did not [18,19,20,21]. As there have been few investigations regarding listening to music in colposcopy with conflicting results, and no studies have been conducted among the Thai population so far, in the current study, the primary endpoint of this study aimed to investigate whether listening to music was effective in relieving pain in colposcopy-directed cervical biopsies. Additionally, the control (no music and headphones) group was added to the comparison.

## 2. Material and Methods

This study was a randomized, controlled trial carried out in the Colposcopy Clinic of Chiang Mai University Hospital. The Faculty of Medicine Research Ethics Committee approved this study with the study code OBG-2563-07089 before its commencement. The trial was registered with the Thai Clinical Trials Registry: TCTR2020716004. Women with abnormal cervical cytology results or positive HPV testing between June 2020 and November 2021 were enrolled. The participating women’s written informed consent was obtained from all attendants. Inclusion criteria were an age between 18 and 60 years and the ability to give written informed consent. Exclusion criteria were being hearing-impaired, pregnancy, previous gynecologic oncology treatment, bleeding disorders, genital tract infection, severe medical conditions, and an inability to communicate in Thai. Demographic and clinical data were collected, including age, parity, menopausal status, history of dysmenorrhea, history of dyspareunia, participant-reported anxiety, history of pelvic inflammatory disease (PID), human immunodeficiency virus (HIV) status, cervical cytology results, human papillomavirus (HPV) testing results, and final histology results.

The participating women were randomly allocated using a computer-generated randomization program divided into three groups. Allocation lists were prepared with sequentially opaque, sealed envelopes containing intervention allocation by the nurses not directly involved with this research and were opened by research assistants. Both the attendants and the research assistants were blinded to the designated allocation. Music was played with mobile headphones. All colposcopic procedures were performed by gynecologic oncologists or gynecologic oncology fellows using a uniform conventional technique. Group 1 (the music group) listened to music through headphones during CDB. They could choose their favorite music and adjust the volume of music by themselves. The other two groups were experimental groups in which the participants in Group 2 wore headphones with no music playing. For the participants in Group 3 (control group), the participants underwent CDB with neither wearing headphones nor listening to music. Colposcopic examination steps were similar among the three groups. All participants were placed in a lithotomy position. A sterile bivalve speculum was inserted to examine the vagina and the cervix, which were used with 5% acetic acid solution. The cervical biopsies were taken from the abnormal acetowhite area. Additional procedures including ECB, ECC, endometrial aspiration biopsy, or cervical polypectomy were performed as per indication and at the discretion of the attending colposcopists. Hemostasis was routinely stopped using Monsel’s solution. After the colposcopic examination, the participating women were observed for 30 min before being discharged home. The research assistants assessed each case to a 10 cm visual analog scale (VAS) at different points of procedural pain during the colposcopic examination. These comprised baseline pain scores (immediately after speculum insertion), biopsy pain scores (immediately after cervical biopsy), and post-procedure pain scores (at 15 min after the procedure). Participants were advised that a score of zero indicated “no pain” and a score of 10 indicated “worst pain”. The primary endpoint was the biopsy pain score. The number of cervical biopsies and the other procedures was also noted. Demographic and clinical information was obtained from medical records.

The primary endpoint was the biopsy pain score. We reasoned that a 1 cm difference in the VAS pain scores between the study groups was the smallest effect that would be clinically meaningful. The effect size (d) of the T-test [22] was calculated from the mean difference (δ) divided by the standard deviation of both groups (SDpooled) = (5.03–3.32) 2.54 = 0.67. Based on significance level (α) = 0.05, power of the test (1 − β) = 0.8, number of groups (k) = 3, and effect size = 0.27, the sample size calculation revealed that 240 participants were essential to obtain adequate power of the test of 80% for detecting the biopsy pain score difference between the three groups. Since the data of this study were normally distributed, the one-way ANOVA test was used to compare the mean pain scores and other continuous variables. Multiple comparisons between each group were performed using the Bonferroni test. The *p*-value of <0.05 was considered statistically significant. All data analyses were conducted using Stata^®^ program version 15 (StataCorp LP, College Station, TX, USA).

## 3. Results

A flow diagram for participants is shown in Figure 1. There were no dropouts for this investigation. Of the 240 women, 80 were randomly designated to the music group (music and headphones; Group 1), 80 to the headphones without music group (Group 2), and 80 to the no intervention group (no music or headphones; Group 3) as the control group. Participants in the three study groups were similar in terms of age, parity, history of dysmenorrhea, menopausal status, anxiety, underlying disease, HIV status, cytology results, and HPV testing results. Atypical squamous cells of undetermined significance (ASC-US) and low-grade squamous intraepithelial lesions (LSILs) were the most common abnormal cervical pathology results among the three groups. In each study group, approximately half of the final histology was normal cervical epithelium and chronic cervicitis. A low-grade squamous intraepithelial lesion was recorded in approximately one-fourth of the participants. Most participants had at least two biopsies (Table 1).

Table 2 shows different stages of pain scores of the procedure. The baseline, the biopsy, and the post-procedure pain scores were comparable among the three groups. The mean baseline pain score between each group was not significantly different (2.83, 95% CI = 2.37–3.28 in Group 1; 2.54, 95% CI = 2.08–3.00 in Group 2; and 2.94, 95% CI = 2.45–3.04 in Group 3, *p* = 0.47). There were also no significant differences in the mean biopsy pain score between each group (4.21, 95% CI = 3.67–4.78 in Group 1; 4.24, 95% CI = 3.67–4.81 in Group 2; and 4.30, 95% CI = 3.74–4.86 in Group 3, *p* = 0.98). The difference in pain score changes from baseline pain to biopsy pain was not statistically significant among the groups (1.39, 95% CI = 0.69–2.08 in Group 1; 1.70, 95% CI = 1.13–2.27 in Group 2; and 1.36, 95% CI = 0.81–1.90 in Group 3, *p* = 0.69). No significant differences in the mean post-procedural pain score among the groups were found (2.34, 95% CI = 1.91–2.77 in Group 1; 2.44, 95% CI = 2.00–2.87 in Group 2; and 2.25, 95% CI = 1.82–2.68 in Group 3, *p* = 0.83).

In terms of multiple comparison analyses (Table 3), the difference changes in pain score from biopsy pain to baseline pain and post-procedural pain score to baseline pain score between each group were not statistically significant. No complications occurred in any of the participants. Figure 2 shows a boxplot of pain scores for baseline, procedure, and post-procedure between study groups.

## 4. Discussion

We did not observe any beneficial effects in lowering procedure-related pain scores in women who listened to music compared to the control group. The comparable time-point pain scores and changes in biopsy pain scores from baseline between the headphones without music group and the control group imply that the placebo effect had no significant interfering role on the outcome assessment and any pain lowering effect observed would essentially result from the music delivered through the headphones.

Despite being a minimally invasive procedure, CDB can cause significant pain and discomfort in a considerable proportion of patients. The sensation of pain in the cervix is transmitted to the brain via pelvic splanchnic nerves carried through lateral parametria. Several studies documented that the CDB procedure was related to different degrees of pain varying from mild to severe. It should also be recorded that pain related to the colposcopy procedure can arise in several steps, involving speculum insertion (dull discomfort), acetic acid application (burning sensation), CDB (pressure or cramping), and extensional investigation such as ECB, ECC, endometrial sampling, and cervical polypectomy. Recently, the ASCCP recommended that at least two and up to four cervical punch biopsies should be taken from abnormal areas found during colposcopic examination [6]. From all these considerations, clinicians should be aware of the usually unrealized suffering and pain occurring in women undergoing colposcopic procedures.

This study was a randomized trial investigating the outcome of listening to music on pain reduction during colposcopy-directed cervical biopsies. Previous randomized, controlled trials performed by Chan et al., Angioli et al., and Law et al. [16,23,24] revealed that listening to music could significantly decrease pain sensations during gynecologic procedures. Pain impulse is a multiplex and unsatisfied sensory that is normally involved with tissue injury [25]. Tissue trauma conducts the release of inflammatory substances with the consequence of nociceptor perception. Pain sensations are then transported to the dorsal horn of the spinal cord, where these connect to the second-order neurons that go to the opposite side of the spinal cord and arise to the spinothalamic tract to the reticular activating system (RAS) and thalamus. The perception of pain sensation is reported in the somatosensory cortex [26]. The first report shows that satisfying emotions when listening to music is involved with dopamine activity in the mesolimbic system, inclusive of both dorsal and ventral striatum [27]. The second report specifies that listening to music can influence cognitive distraction during noxious perception, increasing impulse in the periaqueductal gray area of the midbrain, the region involved with the inhibition of sensory impulse of noxious signals [25,28]. A meta-analysis regarding the effect of listening to music on pain showed that music methods have statistically significant consequences in reducing pain on 0–10 pain scores [29].

However, other randomized, controlled trials found no significant differences in pain, anxiety, or satisfaction between the music-listening group and the control group [7,8,9,19,20]. These findings were consistent with this current trial. Danhauer et al. and Chantawong et al. suggested that the negative findings in their studies, which were inconsistent with the three previously mentioned trials, could be at least partly explained by the limited choice of music genres. However, a combined approach in a prospective way, such as virtual reality or hypnosis, might be another method to decrease pain and anxiety [30,31,32,33,34]. Nevertheless, according to a systematic study, the reduction in pain intensity during the procedure was similar to previous studies in which the participants could select the type of music by themselves. For this trial, there were some possible explanations for the lack of beneficial effects associated with listening to music. First, listening to music through a stereo headset during the procedure could have introduced stress because the participants might have found it difficult to hear what the physician was saying or what was happening. The biopsy performed without verbal warning may have led to increased patient anticipation and excitement, leading to increased pain. Furthermore, some participants could not decide on their preferred music due to their anxiety and excitement at the time of the colposcopy, and therefore, the music selected by the investigator might not have been the participants’ favorite and may not have reduced the pain score in this group.

The strength of the study was the fact that it was a prospective, randomized, controlled trial with an adequate sample size for the outcome of interest and allocation concealment. Colposcopic examinations were performed using a uniform technique with the same level of practitioners. All participants provided information for the final analysis of the important outcomes in the designed study groups without dropouts. Moreover, the placebo group (headphones without music) showed no placebo effect of wearing the headphones, which canceled the surrounding noise. Additionally, the participants were able to choose their preferred music and adjust the volume of music by themselves. However, certain limitations existed. The attendants’ blinding method was only partial, as those in the no intervention group could not be blinded. Furthermore, the blinding of the clinician would not have been practical in this situation. The other limitations of our study also included the fact that participating women wearing headphones may have found it hard to hear what the colposcopist was saying, and the incapacity to know what was happening may have been uncomfortable. For further studies, the most important aim is to find a way to communicate with the patient, especially to let the patient know what is going to happen during the procedure.

## 5. Conclusions

The application of listening to music has no beneficial effect in reducing pain associated with colposcopy-directed cervical biopsy. However, the intervention is safe and may be reasonably considered in overly anxious women undergoing CDB.

## Figures and Tables

**Figure 1 medicina-58-00429-f001:**
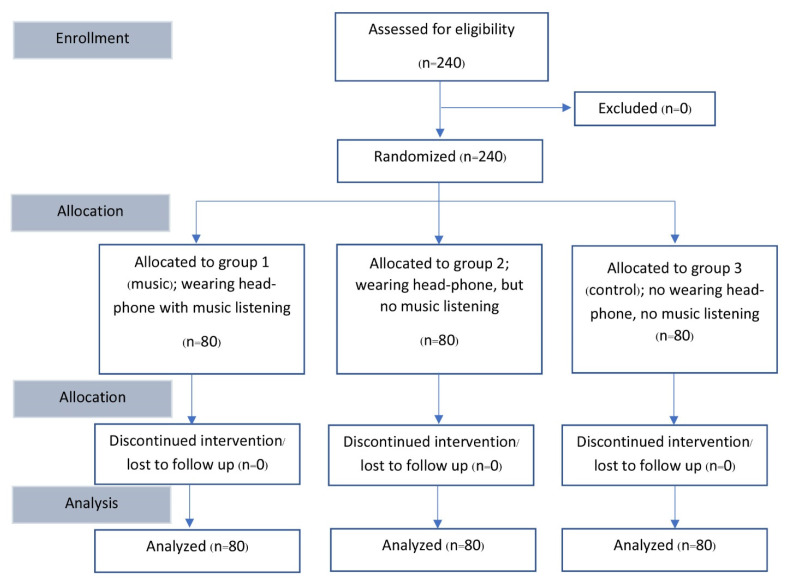
Flow diagram for participating women.

**Figure 2 medicina-58-00429-f002:**
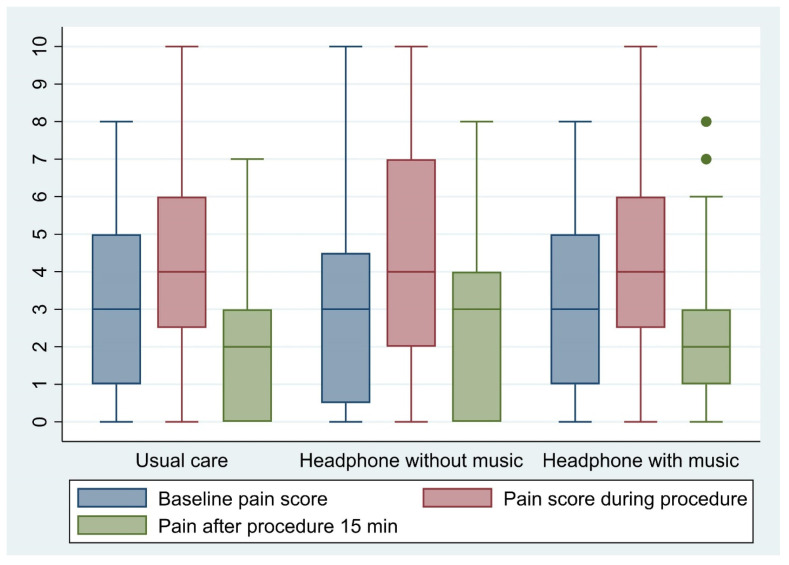
Boxplot of pain scores for baseline, procedure, and post-procedure between study groups.

**Table 1 medicina-58-00429-t001:** Participant characteristics.

Characteristics	Headphones with Music (*n* = 80)	Headphones without Music (*n* = 80)	Usual Care (*n* = 80)	*p*-Value
Age (years)	41.00 (30.00–51.00)	40.00 (32.00–52.00)	43.00 (34.50–52.50)	0.38
Parity				0.07
Nulliparity	35 (43.8)	21 (26.3)	28 (35.0)	
Multiparity	45 (56.3)	59 (73.8)	52 (65.0)	
Menopause status				0.98
No	59 (73.8)	57 (71.3)	58 (72.5)	
Yes	21 (26.3)	23 (28.8)	22 (27.5)	
Dysmenorrhea				0.06
No	43 (53.8)	49 (61.3)	59 (73.8)	
Mild	26 (32.5)	21 (26.3)	14 (17.5)	
Moderate	6 (7.5)	8 (10.0)	7 (8.8)	
Severe	5 (6.3)	2 (2.5)	0 (0.0)	
HIV				0.16
Negative	78 (97.5)	72 (90.0)	75 (93.8)	
Positive	2 (2.5)	8 (10.0)	5 (6.3)	
Underlying disease				0.11
No	63 (78.8)	56 (70.0)	59 (73.8)	
Yes	17 (21.3)	24 (30)	21 (26.3)	
Anxiety score				0.80
No	22 (27.5)	14 (17.5)	15 (18.8)	
Mild	33 (41.3)	38 (47.5)	35 (43.8)	
Moderate	15 (18.8)	18 (22.5)	18 (22.5)	
Severe	10 (12.5)	10 (12.5)	12 (15.0)	
HPV test				0.97
Not done	59 (73.8)	61 (76.3)	58 (72.5)	
Negative	4 (5.0)	5 (6.3)	4 (5.0)	
High-risk HPV-positive	9 (11.3)	9 (11.3)	9 (11.3)	
Other 12 HR HPV-positive	8 (10.0)	5 (6.3)	9 (11.3)	
Cytology				0.14
Normal	6 (7.9)	3 (3.8)	3 (3.8)	
ASCUS	34 (42.5)	24 (30.0)	31 (38.8)	
LSIL	29 (36.3)	33 (41.3)	29 (36.3)	
HSIL	6 (7.5)	14 (17.5)	10 (12.5)	
ASC-H	2 (2.5)	3 (3.8)	7 (8.8)	
No cytology	3 (3.8)	3 (3.8)	0 (0.0)	
Final histology				0.01 *
Normal/inflammation	47 (58.8)	40 (50.0)	39 (48.8)	
LSIL	23 (28.8)	16 (20.0)	29 (36.6)	
HSIL	7 (8.8)	23 (28.8)	11 (13.8)	
Cancer	3 (3.8)	1 (1.3)	1 (1.3)	
Number of biopsies				0.09
1	15 (18.8)	26 (32.5)	15 (18.8)	
2	63 (78.8)	54 (67.5)	64 (80.0)	
3	2 (2.5)	0 (0.0)	1 (1.3)	

* Statistical significance at 0.05.

**Table 2 medicina-58-00429-t002:** Pain scores in the three study groups.

Pain Score	Total (n = 240)	Headphone with Music (n = 80)	Headphones without Music (n = 80)	Usual Care (n = 80)	*p*-Value
Baseline	2.77 (2.50–3.04)	2.83 (2.37–3.28)	2.54 (2.08–3.00)	2.94 (2.45–3.43)	0.47
Biopsy	4.25 (3.92–4.58)	4.21 (3.64–4.78)	4.24 (3.67–4.81)	4.30 (3.74–4.86)	0.98
Biopsy to baseline change	1.48 (1.13–1.83)	1.39 (0.69–2.08)	1.70 (1.13–2.27)	1.36 (0.81–1.90)	0.69
Post-procedure	2.34 (2.09–2.59)	2.34 (1.91–2.77)	2.44 (2.00–2.87)	2.25 (1.82–2.68)	0.83

Pain score was reported by mean (95% Confidence Interval) and compared using ANOVA.

**Table 3 medicina-58-00429-t003:** Multiple comparisons of the pain scores between the study groups.

Variables	Mean Difference (95%CI)	*p*-Value
Biopsy to baseline difference (scores)		
Headphone without music vs. control group	−0.34 (−1.21–0.52)	0.44
Headphone with music vs. control group	−0.03 (−0.90–0.83)	0.94
Headphone with music vs. without music	0.31 (−0.55–1.18)	0.48
Postprocedure to baseline difference (scores)		
Headphone without music vs. control group	−0.59 (−1.33–0.15)	0.12
Headphone with music vs. control group	−0.21 (−0.95–0.53)	0.58
Headphone with music vs. without music	0.39 (−0.35–1.13)	0.30

Bonferroni test was used for multiple comparisons.

## Data Availability

Not applicable.

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
