# Peer review of "Efficacy of Listening to Music on Pain Reduction during Colposcopy-Directed Cervical Biopsy: A Randomized, Controlled Trial"

_medicina, 2022, doi:10.3390/medicina58030429_

Round 1
Reviewer 1 Report
One of the aspects asked about in the demographic questionnaire is the previous presence of self-reported anxiety. Given the obvious subjectivity of this aspect, why was the presence of trait and/or state anxiety not assessed more objectively in order to correlate it with the results obtained?
The health professionals who carried out the examinations were aware of the study; could this not be considered a bias that seriously threatens the validity of the study since they knew to which group each of the participants belonged?
The socio-demographic characteristics and previous pathologies are similar in the three groups. Is this a coincidence? Given the sample size, this fact deserves a more detailed explanation.
What is the main hypothesis proposed by the authors to explain the difference between their results and previous studies?
What alternative proposal to music could be suggested according to the evidence to achieve the desired effects with the use of music?
Author Response
Reviewers’ comment to the manuscript [Medicina-1584986]
entitled “Efficacy of music listening on pain reduction during colposcopy- directed cervical biopsy: A randomized controlled trial”
Dear Editor and reviewers,
We would like to thank you for the opportunity to revise our manuscript to be qualified for publication in Medicina. We have revised and modified some parts of our manuscript as addressed in response to reviewers’ comments. We hope that our responses and revisions would substantially improve the quality of our manuscript and would be qualified for publication in the journal. If there were any further questions or minor points to be addressed or elaborated, please let us know. We would be more than eager to make any further revision.
Reviewer #1:
1. One of the aspects asked about in the demographic questionnaire is the previous presence of self-reported anxiety. Given the obvious subjectivity of this aspect, why was the presence of trait and/or state anxiety not assessed more objectively in order to correlate it with the results obtained?
Response to reviewer#
Thank you for your suggestion. The primary endpoint of this study was the biopsy pain score. The anxiety condition could be simply performed with the aim of investigating the level of anxiety based on the perceptions of the participants during the examination. The perception of anxiety in an individual may be an effect on pain scores. Therefore, the questions and answers in the study are more subjective than objective. From our study in table 1, the anxiety scores of the three group studies are not a significant difference (P = 0.80).
- The health professionals who carried out the examinations were aware of the study; could this not be considered a bias that seriously threatens the validity of the study since they knew to which group each of the participants belonged?
Response to reviewer#
We mention this point in our limitation of the study in lines 252-254.
“However, certain limitations existed. The attendants’ blinding method was only partial as those in the no-intervention group could not be blinded. Furthermore, the blinding of the clinician would not be practical in this situation”.
- The socio-demographic characteristics and previous pathologies are similar in the three groups. Is this a coincidence? Given the sample size, this fact deserves a more detailed explanation.
Response to reviewer#
The sample size was conducted using G power, which was given the effect size from one-way ANOVA according to the objective of this study. The related parameter of sample size calculation consists of the effect size of 0.27, based on these values, and of 0.05 (type I error), and power of 0.8, the minimum sample size required 71 cases per group with a drop-out rate of 20%.
-This study the participants were randomly assigned into three 3 groups equally by a computer-generated randomization program, resulting in a normal distribution (we test with Shapiro- Francia test). This might cause socio-demographic characteristics and previous pathologies similar in the 3 groups.
- What is the main hypothesis proposed by the authors to explain the difference between their results and previous studies?
Response to reviewer#
- The primary endpoint was the biopsy pain score. We reasoned that a 1-cm difference in the VAS pain scores between the study groups was the smallest effect that would be clinically meaningful.
- It has been documented in the literature that the CDB procedure is associated with various degrees of pain ranging from mild to severe. From our previous studies, we could not demonstrate the clinically meaningful significant method to reduce pain in colposcopy-directed biopsy with xylocaine spray.
- According to the previous study [12,20,21] in which a finding has shown that listening to music could reduce pain. So we thought of a music listening method to reduce pain in CDB.
- We send the previous studies of our institute.
Previous studies:
- Lidocaine spray compared with submucosal injection for reducing pain during loop electrosurgical excision procedure: a randomized controlled trial.
Vanichtantikul A, Charoenkwan K.Obstet Gynecol. 2013 Sep;122(3):553-7.
- Effect of lidocaine spray during colposcopy-directed cervical biopsy: A randomized controlled trial.
Wongluecha T, Tantipalakorn C, Charoenkwan K, Srisomboon J.J Obstet Gynaecol Res. 2017 Sep;43(9):1460-1464.
- Effects of lidocaine spray for reducing pain during endometrial aspiration biopsy: A randomized controlled trial.
Piyawetchakarn R, Charoenkwan K.J Obstet Gynaecol Res. 2019 May;45(5):987-993.
- What alternative proposal to music could be suggested according to the evidence to achieve the desired effects with the use of music?
Response to reviewer#
In our study, we use the favorite song for the patient but the other studies used Mozart music, romance music, or classical music.

Reviewer 2 Report
Reviewer Comments
Thank you very much for the opportunity to review the manuscript submission entitled: "Efficacy of Music Listening on Pain Reduction during Colposcopy- Directed Cervical Biopsy: A Randomized Controlled Trial".
The current paper aims to investigate the efficacy of music listening on pain reduction during the colposcopy-directed cervical biopsy. The data is interesting, and it has a relevant rationale.
General comments
The manuscript requires a thorough language revision in English. Some of the sentences are hard to understand.
A significant amount of language in the introduction and discussion need support from published literature. Include citations.
Specific comments
Title: Good
Abstract
- Include the mean age of the participants in the abstract
- Include MeSH terms as keywords
Introduction
- Include references for the sentences in the introduction.
- "The manuscript requires a thorough language revision in English. Some of the sentences are hard to understand." – requires reference
- In addition, cervical cancer was the fourth leading cause of mortality in female cancer. - requires reference
- In Thailand, cervical cancer is the third most common cancer, ranking after breast and colorectal cancer. There were 9,157 new cervical cancer cases in Thai women, or 25 cervical cancer cases were diagnosed every day. - requires reference
- Need more emphasis on Scientific background and explanation of rationale.
- There are so many techniques and modalities that are proven to reduce pain. So what is the rationale for selecting music therapy to decrease pain? Need a strong explanation.
- Provide a hypothesis of your study
Methods
- Institutional Review Board (IRB) reference number needs to be included in the manuscript.
- Inclusion and exclusion criteria need more elaboration
- Clearly mention who generated the random allocation sequence, who enrolled participants, and who assigned participants to interventions
Discussion
- Give a cautious overall interpretation of results considering objectives, limitations, the multiplicity of analyses, results from similar studies.
- Discuss the generalizability of the trial findings

Author Response
Reviewers’ comment to the manuscript [Medicina-1584986]
entitled “Efficacy of music listening on pain reduction during colposcopy- directed cervical biopsy: A randomized controlled trial”
Dear Editor and reviewers,
We would like to thank you for the opportunity to revise our manuscript to be qualified for publication in Medicina. We have revised and modified some parts of our manuscript as addressed in response to reviewers’ comments. We hope that our responses and revisions would substantially improve the quality of our manuscript and would be qualified for publication in the journal. If there were any further questions or minor points to be addressed or elaborated, please let us know. We would be more than eager to make any further revision.
Reviewer #2:
1. The manuscript requires a thorough language revision in English. Some of the sentences are hard to understand
Response to reviewer#2:
- We consulted an academic English editor to revise the manuscript.
- We updated some of the references as suggested.
- A significant amount of language in the introduction and discussion need support from published literature. Include citations
"The manuscript requires a thorough language revision in English. Some of the sentences are hard to understand." – requires reference
In addition, cervical cancer was the fourth leading cause of mortality in female cancer. - requires reference
In Thailand, cervical cancer is the third most common cancer, ranking after breast and colorectal cancer. There were 9,157 new cervical cancer cases in Thai women, or 25 cervical cancer cases were diagnosed every day. - requires reference
- Response to reviewer#2:
Thank you for your suggestion.
- We modified some parts of the introduction and add more citation in the manuscript.
Introduction: line 50 -51.
In developing countries of female cancer, cervical cancer was the second most common of cancer and the third most common cause of cancer death [2-4].
References:
- Bray F, Ferlay J, Soerjomataram I, Siegel RL, Torre LA, Jemal A. Global cancer statistics 2018: GLOBOCAN estimates of incidence and mortality worldwide for 36 cancers in 185 countries. CA Cancer J Clin. 2018;68(6):394-424.
- Torre LA, Bray F, Siegel RL, Ferlay J, Lortet-Tieulent J, Jemal A. Global cancer statistics, 2012. CA Cancer J Clin. 2015;65(2):87-108.
- Human Papillomavirus and Related Cancers in the World. Summary Report 2010. [Internet]. 2010 [cited February 1,2021]. Available from:http://www.who.int/hpvcentre/en/.
Introduction: line 66 - 68
Previous studies trying to find the methods to reduce the pain and discomfort during colposcopy to improve treatment outcomes such as lidocaine injection, lidocaine spray, or forced coughing [8-11].
References:
- Karaman E, Kolusarı A, Alkış Ä°, Çetin O. Comparison of topical lidocaine spray with forced coughing in pain relief during colposcopic biopsy procedure: a randomised trial. J Obstet Gynaecol. 2019;39(4):534-8.
- Naki MM, Api O, Acioglu HC, Uzun MG, Kars B, Unal O. Analgesic efficacy of forced coughing versus local anesthesia during cervical punch biopsy. Gynecol Obstet Invest. 2011;72(1):5-9.
- Oyama IA, Wakabayashi MT, Frattarelli LC, Kessel B. Local anesthetic reduces the pain of colposcopic biopsies: a randomized trial. Am J Obstet Gynecol. 2003;188(5):1164-5.
- Öz M, Korkmaz E, Cetinkaya N, BaÅŸ S, Özdal B, Meydanl MM, et al. Comparison of Topical Lidocaine Spray With Placebo for Pain Relief in Colposcopic Procedures: A Randomized, Placebo-Controlled, Double-Blind Study. J Low Genit Tract Dis. 2015;19(3):212-4.
- Need more emphasis on scientific background and explanation of rationale.
There are so many techniques and modalities that are proven to reduce pain. So what is the rationale for selecting music therapy to decrease pain? Need a strong explanation.
Provide a hypothesis of your study
- Response to reviewer#2:
- From our previous study with xylocaine spay in the colposcopy-directed cervical biopsy. We could not show the clinical significant benefit of xylocaine spray in our study. Given the previously known music therapy, there was further interest in the mechanism of anxiety and pain reduction associated with music listening. In addition, listening to favorite music can stimulate the release of dopamine and induce cognitive distraction from noxious stimuli. It is also an easy tool that can be done anywhere and anytime. Lastly, as its low cost, it is suitable for application into practice. And from the previous study, there have been some studies with have conflict in result, as well as, no study in Thai population. So, I would like to study the effect of music listening on pain reduction during CDB in our setting.
- Methods
Institutional Review Board (IRB) reference number needs to be included in the manuscript.
Inclusion and exclusion criteria need more elaboration
Clearly mention who generated the random allocation sequence, who enrolled participants, and who assigned participants to interventions
- Response to reviewer#2:
Thank you for your suggestion. We insert the Institutional Review Board (IRB) reference number in line 80.
- The Faculty of Medicine Research Ethics Committee approved this study with study code: OBG-2563-07089
- The researcher used a computer-generated randomization program to divide the participants into three groups
- The participants were enrolled assigned participants to interventions by researcher assistants.
- We modified the inclusion and exclusion criteria in lines 84-86.
Inclusion criteria were age between 18 and 60 years, ability to give written informed consent. Exclusion criteria were hearing-impaired, pregnancy, previous gynecologic oncology treatment, bleeding disorders, genital tract infection, severe medical conditions, and inability to communicate in Thai.
- Give a cautious overall interpretation of results considering objectives, limitations, the multiplicity of analyses, results from similar studies.
Discuss the generalizability of the trial findings
- Response to reviewer#2:
Thank you for your suggestion. We mention in the discussion section on line 252 – 258.
“However, certain limitations existed. The attendants’ blinding method was only partial as those in the no-intervention group could not be blinded. Furthermore, the blinding of the clinician would not be practical in this situation. The other limitations of our study were also found with participating women wearing head-phones may have found it hard to hear what the colposcopist was saying, and incapacity to know what is going on may be uncomfortable. For further studies, the most important point is to find a way to communicate with the patient, especially to let the patient know what we are going to do during the procedure.”

Round 2
Reviewer 1 Report
The authors have adequately justified the modifications introduced in response to the reviewer's requirements, and the text is therefore considered suitable for publication.
Author Response
+Reviewers’ comment to the manuscript [Medicina-1584986]
entitled “Efficacy of music listening on pain reduction during colposcopy- directed cervical biopsy: A randomized controlled trial”
Dear Editor and reviewers,
We would like to thank you for the opportunity to revise our manuscript to be qualified for publication in Medicina. We have revised and modified some parts of our manuscript as addressed in response to reviewers’ comments. We hope that our responses and revisions would substantially improve the quality of our manuscript and would be qualified for publication in the journal. If there were any further questions or minor points to be addressed or elaborated, please let us know.
Reviewer #1:
The authors have substantially improve the quality of the manuscript helped by the reviewers comments. Some concerns should be considered before publication.
- Introduction section:
The rational of the study should be reinforce, notably by adding some references and by clearly indicating hypothesis of the work related to literature.
Music has been already used in other pathologies than colposcopy. You have to add a sentence to provide evidence that music therapy should continue to be investigated.
For example:
Music for pain relief.
Cepeda MS, Carr DB, Lau J, Alvarez H.
Cochrane Database Syst Rev. 2006 Apr 19;(2):CD004843. doi: 10.1002/14651858.CD004843.pub2.
PMID: 16625614
The effectiveness of music as an intervention for hospital patients: a systematic review.
Evans D.
J Adv Nurs. 2002 Jan;37(1):8-18. doi: 10.1046/j.1365-2648.2002.02052.x.
PMID: 11784393
- Second paragraph of the introduction:
“Several studies.. colposcopy” Please add some references at the end of this sentence.
“Some studies…, whereas the others did not” Please add some references at the end of the sentence.
“As there have been few types of research” please clarify this sentence. What is ‘few types of research’?
At the end of the paragraph please indicate primary and secondary objectives, and hypothesis.
# Response to reviewer
Thank you for your suggestions.
Introduction section:
We add the sentence and add the references “Music has been investigated in chronic pain, pain reduction of children, and other procedures like orthopedic surgery or cystoscopy with promising outcomes [12-15]. “ in the second paragraph of the 7th sentence.
Gauba A, Ramachandra MN, Saraogi M, Geraghty R, Hameed BMZ, Abumarzouk O, Somani BK. Music reduces patient-reported pain and anxiety and should be routinely offered during flexible cystoscopy: Outcomes of a systematic review. Arab J Urol. 2021 Mar 3;19(4):480-487. doi: 10.1080/2090598X.2021.1894814. eCollection 2021.
PMID: 34881066
2. Ting B, Tsai CL, Hsu WT, Shen ML, Tseng PT, Chen DT, Su KP, Jingling L. Music Intervention for Pain Control in the Pediatric Population: A Systematic Review and Meta-Analysis.
J Clin Med. 2022 Feb 14;11(4):991. doi: 10.3390/jcm11040991.
PMID: 35207263
3. Hsu HF, Chen KM, Belcastro F. The effect of music interventions on chronic pain experienced by older adults: A systematic review.
Effect of Music Therapy on Pain, Anxiety, and Use of Opioids Among Patients Underwent Orthopedic Surgery: A Systematic Review and Meta-Analysis.J Nurs Scholarsh. 2022 Jan;54(1):64-71. doi: 10.1111/jnu.12712. Epub 2021 Nov 5.
PMID: 34741407.- Patiyal N, Kalyani V, Mishra R, Kataria N, Sharma S, Parashar A, Kumari P. Effect of Music Therapy on Pain, Anxiety, and Use of Opioids Among Patients Underwent Orthopedic Surgery: A System-atic Review and Meta-Analysis.
Cureus. 2021 Sep 29;13(9):e18377. doi: 10.7759/cureus.18377. eCollection 2021 Sep.
PMID: 34725621
- We add the references at the end of this sentence “Several studies.. colposcopy” .
- We add the references at the end of this sentence “Some studies…, whereas the others did not”.
- We add primary endpoint of the study at the end of the second paragraph.
Methods section:
Paragraph 4: “we reasoned that a 1-cm difference in the VAS… would be clinically meanful”. How the author choose this cutoff? There is some recommendations in the literature, and they clearly indicate that minimal clinical of change is 2-cm or 30% in the VAS score (Dworkin, R.H., Turk, D.C., Wyrwich, K.W., Beaton, D., Cleeland, C.S., Farrar, J.T., Haythornthwaite, J.A., Jensen, M.P., Kerns, R.D., Ader, D.N., Brandenburg, N., Burke, L.B., Cella, D., Chandler, J., Cowan, P., Dimitrova, R., Dionne, R., Hertz, S., Jadad, A.R., Katz, N.P., Kehlet, H., Kramer, L.D., Manning, D.C., McCormick, C., McDermott, M.P., McQuay, H.J., Patel, S., Porter, L., Quessy, S., Rappaport, B.A., Rauschkolb, C., Revicki, D.A., Rothman, M., Schmader, K.E., Stacey, B.R., Stauffer, J.W., von Stein, T., White, R.E., Witter, J., Zavisic, S., 2008. Interpreting the clinical importance of treatment outcomes in chronic pain clinical trials: IMMPACT recommendations. J. Pain. Feb;9(2), :105–21. https://doi.org/10.1016/j. jpain.2007.09.005.)
In addition, please indicate the outcomes assessed 0 to 10.
# Response to reviewer:
- We reasoned that a 1-cm difference in the VAS pain scores between the study groups was the smallest effect that would be clinically meaningful. The effect size (d) of the T-test was calculated from the mean difference (δ) divided by the standard deviation of both groups (SDpooled) = (5.03 – 3.32) 2.54 = 0.67. Based on significance level (α) = 0.05, power of the test (1-β) = 0.8, number of groups (k) = 3, and effect size = 0.27, the sample size calculation revealed that 240 participants were essential to obtain adequate power of the test of 80% for detecting the biopsy pain score difference between the three groups. We understand that a 1-cm difference in the VAS pain scores will need more sample size than a 2-cm difference in the VAS pain scores. From our study, there were no significant differences between each group in terms of mean biopsy pain score with the 1-cm difference in the VAS pain scores. Therefore, the ability to detect the difference between 3 groups with a 1-cm difference or a 2-cm difference will no effect on our outcomes.
- The visual analog scale consisted of a 10-cm line scaled from 0 (no pain) to 10 (worst possible pain). The primary endpoint was the biopsy pain score. We mentioned in material and methods section in the second paragraph.
- Results section:
Please separate group 1, group 2 and group 3 results by a ‘;’
For example: (2.83, 95% CI = 2.37-3.28 in Group 1; 2.54, 95% CI = 2.08 -3.00 in Group 2; and 2.94, 95% CI = 2.45-3.04 in Group 3, P = 0.47).
Please indicate if there is a main effect of time whatever the group. This is an important issue than can showed an effect of the intervention without any difference between groups and no interaction (from your ANOVA analysis).
# Response to reviewer:
- Thank you for your suggestion. We separate group 1, group 2, and group 3 results by a ‘;’ in the results section.
- Additional information, analysis mixed-ANOVA for the main effect of group, time, and the interaction effect between group and time. The results in the table below (Appendix 1) show the time effect is only a significant difference (p=<0.001), in which pain score on biopsy time was higher than both baseline and post-procedure. It is reasonable that the patient has some pain after having a biopsy. The group and time interaction effect is shown a non-significant difference with an explained variance of 0.6% (p=0.603). Therefore, time change did not confound the intervention. The group or intervention effect is also a non-significant difference (p=0.435) with controlling the time effect and the interaction effect.
Appendix 1. Results of mixed-ANOVA for pain score comparison between groups and time.
Variable |
Group |
Time |
Group effect |
Time effect |
Interaction between Group and Time effect |
|||||
Baseline |
Biopsy |
Post-procedure |
η2 |
p |
η2 |
p |
η2 |
p |
||
Pain score |
Headphone with music |
2.83 ± 2.09 |
4.21 ± 2.61 |
2.34 ± 1.97 |
0.003 |
0.435 |
0.267 |
<0.001* |
0.006 |
0.603 |
Headphones without music |
2.54 ± 2.09 |
4.24 ± 2.61 |
2.44 ± 1.99 |
|||||||
Usual care |
2.94 ± 2.24 |
4.30 ± 2.57 |
2.25 ± 1.95 |
*p<0.05 (statistical significance)
4.Discussion section:
Second sentence of the 4th paragraph “these findings… music genres.”
The author could be considered combined approach in a perspective way such as Virtual Reality, hypnosis, etc. to decrease pain and anxiety
# Response to reviewer:
Thank you for your suggestion. We add the sentence “However, the combined approach in a prospective way such as virtual reality or hypnosis might be the other method to decrease pain and anxiety.” in the discussion section in the third sentence of the 4th paragraph.
